# Family Members’ Explanatory Models of Cancer Anorexia–Cachexia

**DOI:** 10.3390/healthcare12161610

**Published:** 2024-08-13

**Authors:** Susan McClement

**Affiliations:** Rady Faculty of Health Sciences, College of Nursing, University of Manitoba, Winnipeg, MB R3T 2N2, Canada; susan.mcclement@umanitoba.ca

**Keywords:** cancer cachexia, explanatory models, family care

## Abstract

The experience of bearing witness to the lack of appetite and involuntary weight loss that characterizes cancer anorexia–cachexia syndrome (CACS) is reported to be stressful for family members. Research identifies that family members engage in a wide range of behaviors in response to a relative who shows minimal interest in eating and is literally ‘wasting away’ before their eyes. Some families, though concerned about the symptoms of CACS, do not dwell excessively on the patient’s nutritional intake while others continually harass the patient to eat and petition health care providers for aggressive nutritional interventions to eat in an attempt to stave off further physical deterioration. While studies have detailed how family members respond to a terminally ill relative with CACS, empirical work explicating the explanatory models of CACS that they hold is lacking. Explanatory models (EMs) reflect the beliefs and ideas that families have about why illness and symptoms occur, the extent to which they can be controlled, how they should be treated, and how interventions should be evaluated. To address this gap in the literature, a grounded theory study guided by Kleinman’s Explanatory Model questions was conducted with 25 family members of advanced cancer patients. The core category of ‘Wayfaring’ integrates the key categories of the model and maps onto Kleinman’s questions about CACS onset, etiology, natural course, physiological processes/anatomical structures involved, treatment, and the impacts of disease on patient and family. Findings suggest that a divergence between some biomedical constructions of CACS and explanatory models held by family members may fuel the family–health care provider conflict, thereby providing direction for communication with families about care of the patient with anorexia–cachexia.

## 1. Introduction

Patients with advanced malignancies frequently experience anorexia, involuntary weight loss, muscle wasting, and profound fatigue [1,2,3]. These symptoms are manifestations of the cancer anorexia–cachexia syndrome (CACS); a complex process of metabolic abnormalities resulting in skeletal muscle catabolism that affects upward of 80% of those with advanced disease [4,5]. Clinical consequences of CACS include reduced response to antineoplastic drugs, decreased tolerance to radiation and chemotherapy, poor performance status, and decreased survival [6,7]. The symptoms of CACS in a terminally ill relative are distressing for family members who view them as harbingers of death [8,9,10]. Family members’ concerns are well founded; CACS is predictive of poor prognosis and is estimated to be responsible for up to twenty-five percent of cancer deaths [11,12].

Concerted research efforts to date have not identified any effective pharmacological or nutritional interventions to reverse CACS [13].

Postulated mechanisms for the etiology of cancer cachexia are well described in the biomedical literature [1,2,3,4,5,6,7]. However, previous research suggests that family members often do not understand the complex pathophysiology causing CACS invoking explanations for patient anorexia and weight loss to such factors as patient stubbornness, the quality of hospital food, the volume of medications prescribed, and even health care provider nutritional neglect of the patient [14,15]. Not surprisingly, health care providers report that interactions with family members around the nutritional care of a patient with CACS are among the most conflict-laden and vexing they face [15]. Alternative explanations that families hold about the cause and management of CACS have not been systematically examined or contrasted with prevailing biomedical understandings, constituting a gap in the literature that needs to be addressed. The ability of health care providers to design effective family-based interventions for CACS is predicated on a thorough analysis of families’ CACS-related concerns. While studies have been conducted detailing *what family members do* in response to the symptoms of CACS [14,15,16], empirical work systematically examining *why family members do what they do* is lacking. Thus, research is needed to provide a comprehensive understanding of the ways that family members make sense of CACS in the context of advanced cancer.

One approach to this understanding is explication of the explanatory models (EMs) of CACS that family members hold. EMs are cognitive representations that people have about a particular episode of illness and its symptoms [17]. They help people make sense of illness by providing explanations about when it occurs, its causes, the extent to which it can be controlled, and how it should be managed [18]. As such, they form the basis of behavior. Approaches to explicating cognitive representations of illness reported in the literature include the Common Sense Model of Illness (CSM) [19] and the Illness Perception Questionnaire (IPG) [20]. A limitation of the CSM is that it fails to explore notions of cure or control of symptoms [19]. The IPQ with its fixed range of predetermined causal explanations assumes that the range of beliefs about the etiology of symptoms is already known [20]. In light of these limitations, this study utilized Kleinman’s approach to understanding explanatory models that involves a process of qualitive inquiry that generates rich, multilayered descriptions that highlight the personal meaning of illness and offers explanations that can guide the selection of various therapies [17]. Kleinman advocates for detailed interviews with people to elicit an understanding of their explanatory models regarding etiology; timing and mode of onset; physiological processes and anatomical structure(s) involved; natural course and severity; prognosis, treatment options, and their rationale. Variation in family members’ explanatory models of illness and those articulated in the biomedical literature, while perhaps reflecting some common underlying understandings, may have important dimensional differences that ground family–health care provider conflicts. Research explicating an understanding of family members’ explanatory models of CACS will sensitize clinicians to these potential areas of divergence. This understanding will allow for the development of interventions in response to family members’ unique needs. 

### Research Questions

The specific questions driving this study were: (1) what is the process by which family members arrive at their explanatory models of cancer anorexia cachexia; and (2) what are the points of convergence and divergence between family’s explanatory models and those in the biomedical literature?

## 2. Materials and Methods

Ethics Considerations: Ethical approval to conduct the study was obtained from the University of Manitoba Research Ethics Board. Permission to access the Pain and Symptom Management and Gastrointestinal Oncology Clinics where patients with cachexia and their family members are seen was obtained from St. Boniface Hospital and CancerCare Manitoba. Informed consent was obtained from all subjects involved in the study. No personal identifying information was recorded on data collected for the study. Each participant was assigned a unique code number that was linked with their data. A master list of code numbers and participant names were stored separately from consent forms. The names of participating family members were not shared with clinicians at the recruitment sites. All consents, demographic forms, audio-taped interviews, field notes, and interview transcripts were securely stored. Computer files containing interview transcripts and demographic data were stored on a computer protected by an encrypted password known only to the author and Study Nurses. 

Design: A grounded theory approach was used to address the aims of the study. This qualitative approach results in the construction of a theoretical account of the general features of a topic by grounding the account in empirical data [21,22]. Grounded theory is particularly well suited for studies where the phenomenon of interest is complex, has not been well described, and is socially constructed [23]. Given the dearth of empirical work examining the process by which family members construct their explanatory models of CACS, a grounded theory approach was deemed appropriate for the project. 

Setting, Sampling, and Recruitment: The primary informants for the study were family members of palliative cancer patients with objective characteristics of cachexia being seen at two Pain and Symptom Management clinics and a Gastrointestinal Oncology Clinic at a provincial cancer center that regularly see patients with cachexia and their families. The clinics provide a multidisciplinary assessment of patients in active or palliative phases of care who have symptoms related to their cancer or its treatments that are difficult to resolve. Family members are considered integral to the assessment process and are encouraged to attend the clinics. 

Cachexia was objectively measured as per symptoms on the Patient-Generated Subjective Global Assessment (PG-SGA) [24] of (i) self-reported weight loss of 10% or more in the past six months and a decrease in weight in the past two weeks; (ii) self-reported food intake of less than usual; and (iii) self-report of any of the symptoms that can interfere with eating as outlined on the PG-SGA. The PG-SGA is a validated nutritional screening tool developed for use in patients with cancer, and it is endorsed by the Oncology Nutrition Dietetic Practice Group of the American Dietetic Association as the standard for nutrition assessment of patients with cancer. The tool has sensitivity and specificity of 98% and 82%, respectively, and evidence exists attesting to the reliability of the patient self-reported height, weight, weight history, and patient-perceived level of dietary intake [24]. 

A dyadic recruitment approach was used for the study. First, patients identified with cachexia as per the PG-SGA who were 18 years of age or older with no evidence of cognitive impairment and able to read and speak English were asked by the Study Nurse to provide the name and contact information of a family member who would be able to comment on their illness. Patients were ineligible for participation if they were receiving tube-feeding or other form of artificial nutrition or hydration; were receiving palliative radiation or chemotherapy; refused to complete the PG-SGA or nominate a family member. Two nurses experienced in conducting qualitative interviews supported the work of the study. They first contacted the family member identified by the patient, explained the purpose of the study, and invited participation in it. Family members were eligible to participate if they were 18 years of age or older, with no evidence of cognitive impairment, able to read and speak English, and willing to be interviewed by the Study Nurse. Face-to-face interviews were arranged for family members interested in taking part in the study. 

Data Collection: Demographic information was collected from family members to describe the sample characteristics. Face-to-face audio-recorded interviews were conducted using an interview guide informed by Kleinman’s explanatory model question framework [21], with probes and follow-up questions being posed to confirm and clarify what participants said. Interviews ranged from 45 min to 2 h in length. Consistent with a grounded theory approach, data emerging from initial interviews helped to shape and refine questions that were asked in subsequent interviews. Immediately following an interview, contextual features of the interaction, such as family members’ non-verbal behaviors, affect, and tenor of the interview were recorded by the Study Nurse in field notes. Interviews were transcribed verbatim and reviewed against the audio-recordings by the Study Nurse who conducted the interview to ensure accuracy.

Data Analysis: Demographic data were analyzed using descriptive statistics. Consistent with a grounded theory method, data collection and analysis occurred concurrently. A three-stage process was used [25]. First, open coding was conducted by completing a line-by-line review of interview transcripts to identify, name, categorize, and describe the phenomena found in the text and its properties. Next, axial coding was completed where the codes and their properties were assigned to clusters or categories. Finally, level three coding identified the core category in the data that integrated the major categories.

Well-accepted strategies for ensuring rigor in qualitative research were applied in this study. The credibility or truth-value of the work [26] was addressed through sustained Study Nurse engagement with participants during interviews and a sustained period of data analysis by the author. Dependability, the degree to which a detailed and balanced examination of the phenomena of interest is achieved [27], was addressed through triangulation of recruitment sites (i.e., Pain and Symptom Management clinic and a GI clinic) and data sources (i.e., in-depth interviews and field notes). Confirmability, the extent to which study findings reflect the participants’ experiences and not the biases of the researcher [28], was addressed by keeping a detailed audit trail through processes and decision points made regarding the identification of major findings generated from the data. Transferability, the extent to which the emerging theory reflected the real concern of participants [29], was confirmed through member check with participants to ensure that study findings resonated with them. The Study Nurse also confirmed initial interpretations with participants directly or within interviews as needed. 

## 3. Results

A total of 82 patients were approached to participate in the study, with 29 agreeing to nominate a family member. Reasons provided by the 53 patients who declined included ‘not being interested’, ‘not wanting to burden family members by nominating them to be contacted by the Study Nurse’, ‘not feeling well enough to participate’, and ‘not having living relatives to nominate’. Twenty-five of the twenty-nine family members nominated by the patient agreed to participate in the study. Reasons for declining included ‘not being interested’ and ‘being too busy’. Twenty-one of the family member participants were female and four were male. Of the female participants, 13 were the daughter of the patient and 7 were the patient’s wife or common-law partner. Three of the male family members were the husband of the patient and one was a son. Family members ranged in age from 37 to 62 years. Eight family members had completed a university degree; ten had completed some community college education; seven had completed high school. 

### Core Category: Wayfaring

The core category emerging from explanatory model interviews with family members was that of wayfaring. Wayfaring refers to family members watching, listening, feeling, and responding as they constructed their explanatory models and made passage across the foreign landscape of cancer cachexia in order to make sense of the path unfolding before them. Wayfaring integrates the key categories of making sense of CACS onset, etiology, natural course, physiological processes, and anatomical structures involved, treatment options, and impacts on patient and family. Three sub-categories of Wayfaring were identified: (1) Unbeknownst Wayfaring; (2) Attributional Wayfaring; and (3) Oscillatory Wayfaring. The outcome of wayfaring was not arrival at a final destination but that of adjusting one’s gait based upon reflection and critique of their experience of what was happening with the patient over time. A schematic of the Wayfaring model and its relationship to Kleinman’s Explanatory Model interview questions is depicted in Figure 1.

Unbeknownst Wayfaring: This sub-theme captures the fact that although the patient did not yet have a formal cancer diagnosis, families had, unbeknownst to them, already begun the journey of making sense of what would soon be recognized as symptoms of cancer anorexia–cachexia. Families described subtle but not yet alarming changes with oral intake. They noted patients reported feeling ‘not quite right’, which was frequently attributed to having the flu or having ‘picked up a bug’ somewhere. In unbeknownst wayfaring, families did not yet have knowledge of the cancer diagnosis, and thus did not situate patients’ symptoms within that context. However, they were able to do so in retrospect. The husband of one patient noted:


*She had a month before she was diagnosed, and I was sort of thinking she was a little bit off. She didn’t have much appetite. I mean she still ate what you would probably call three meals, but not very big meals a day. And what she said was strange…that she had no appetite. At that time, I wasn’t connecting things….*


The daughter of this patient said:


*We thought she had the flu, And you know, that can ruin your appetite for a while. But after that, she never did get back to normal in terms of energy, and her appetite was worse after that. That was the real defining time, when, the you know, worst decline started.*


The wife of this patient offered the following explanation:


*He just said he didn’t feel like eating, but you know, you think you have a bug. It is the start of school and getting back into a routine. We both just attributed it to having caught a bug so I would say that we weren’t even really thinking about weight loss at that point.*


When appetite did not improve and weight loss began, family members had to rethink their initial explanations for the patient’s symptoms. At this juncture, families were moving from unbeknownst wayfaring to attributional wayfaring. Looking back, a husband said:


*So what I say is that I now know that it is easy to look at things and say, yup, this is what it is, but when health is being negative, and not just a poor appetite but when you are losing a bit of weight when it becomes dire, like you have to look at any other causes that it might be. You have to look at other causes I think and rule them out.*


Attributional Wayfaring: This sub-theme reflects the explanations families provided to account for the patient’s lack of appetite and involuntary weight loss in the context of a cancer diagnosis. They offered a variety of explanations to account for the anorexia and unintentional weight loss that patients exhibited. Chief among these was caloric deficit caused by inadequate intake of calories and protein due to (i) symptoms caused by the tumor, such as causing a blockage or stealing calories for its own growth; (ii) the side effects of treatment such as nausea/vomiting and altered sense of taste; (iii) pain; (iv) poor fitting dentures; and (v) generalized exhaustion.

In response to the attribution of caloric deficit as a driver of CACS, family member-initiated strategies used to increase caloric intake included the use of trial and error to find something the patient would eat; keeping the cupboard stocked with things that might appeal to the patient, ‘just in case’; sneaking calories in by doubling up on sugar and adding protein powder. A patient’s husband explained:

*We tried lots of the tricks of protein powder and cottage cheese and different things like that. High calorie foods like chocolate covered almonds. We would try to put more fat into things like lots of cream and sugar in her tea. Just to sneak in as many calories as I can*.*I do think that the sheer exhaustion plays into it a little bit. And not having very good tase buds, or her taste being changed is also probably affecting it a bit. You know, not being able to smell the food cooking as well as she used to, and she tastes it, it doesn’t taste like what she would expect it to be*.

In regard to tumor-related attributions of CACS, the daughter of a patient reflected:

*Well somehow, I am imagining that the tumor is interfering with something. I don’t know if it is plugging something or blocking something or the body reacting to something it knows shouldn’t be there. I don’t really understand how that stuff works. Just that things get screwed up somehow from the cancer and it won’t work right and the appetite is one of them*.

The husband of a patient said:

*I think the cancer is obviously fighting his normal system, what is left of it, for those calories and it is just taking it in. Those cells to grow, because the last scan shows that all of the masses have grown and the have to get their energy from somewhere so they are taking in whatever he is taking in*.

In attributing CACS to issues related to side effects of treatment, this patient’s husband asserted that the morphine prescribed for his wife’s pain caused her lack of appetite: 

*Well, I am pretty sure that a lot of it could be contributed to the narcotics she is on. I mean, the narcotics would be an appetite suppressant. I just know that if you have watched any TV at all, you know that junkies would rather have a fix than a sandwich*.

Family members also implicated patients’ emotional responses to explain lack of appetite and weight loss. The wife of a patient reflected:

*I think when he knew it was bone cancer, it hit him then that there was no way out now. There is no question of I’ll get better, death is coming. The whole appetite for life, including food just sort of goes by the wayside when you are figuring that out, I think*.

The husband of this patient reasoned: 

*I am sure she is experiencing some emotional distress and maybe depression and so eating is just not a priority. She has a lot of reasons to be depressed right now so I wouldn’t be surprised at all if that had a large factor to play*.

A daughter of one of the patients believed that in part, the patient’s not eating was a way of communicating the distress they were experiencing about their illness and a desire to be cared for:

*Well, you just have to look at like, the suffragettes who didn’t eat because of protest, I think or Ghandi. You are trying to get a message across and nobody is listening to you. Maybe this was a manifestation of saying, look there really is something wrong with me. He would never articulate to a brother or a son or daughter, I am really afraid of dying and I’m afraid of what is happening to my body. His emotional reaction to this is not eating because it is such a mess of tangled stuff*.

The subconscious desire to be ‘thin’ was implicated by this husband in commenting upon his wife’s weight loss in the context of CACS:


*And I think, the fact that she has been eating less and I think subconsciously somewhere in the back of her mind she is, “I’m fat, I want to lose some weight.”*
*She knew she had to take something in. She didn’t want to….But that was just sheer will power to sit up and actually drink it and it took her about half an hour. She just takes it, mind over matter*.

A positive attitude was an important part of patient mindset strategies identified by a patient’s son:

*But I think your will to eat, the way you look at things, like if your appetite decreases and your weight loss continues then I think your morale gets lower and you have more of a defeatist attitude, and think they go hand in hand with either a positive outlook or a negative outlook with regards to your intake*.

The other main approach families described to help increase caloric intake involved taking an aggressive stance in setting expectations for when and how much the patient would eat. Such a stance was driven by the high stakes of patient survival. This wife was adamant:

*My modus operandi is that you have to eat to stay alive. I have to tell you I was quite militant because I thought he was giving up. And when people stop choosing food, it’s like they are stopping to choose life. I was hysterical. I had a few drama queen moments*.*Well at meal time, it is just forcing her to eat a little bit more. When she is pushing the plate away, you say one more bite, and she turn around and slaps me. It has turned me into a nag. I mean you have to eat to survive you know*.

Families looked to health care providers to initiate strategies that would support patients’ nutritional intake. Chief among these was the effective management of symptoms and side effects of chemo/radiation therapies. Family participants reported that steroid medications were helpful in increasing appetite, at least in the short term. The use of medical marijuana to treat lack of appetite was identified by some oncologists; however, family participants felt that this strategy was offered too late to be of any benefit to the patient. A patient’s wife observed:

*My husband kept asking the oncologist, ‘what can I do to stimulate my appetite? The oncologist said, we can give you a prescription for some synthetic cannabis. We thought that was ridiculous and it was too late to really put any wight back on. He was down to 129 pounds*.

Consultation with a dietician was identified as being helpful by some family participants who appreciated specific direction about increasing protein intake and strategies related to it. This husband noted:

*Since we were put in touch with and speaking to the dietician, I think we have actually managed to keep her calorie and protein intake high enough to match her energy output. She had to hear that you need to get calories and protein in to your body*.

However, as illustrated in this quote by a patient’s wife, some family members had not been referred to a dietician:

*We were told that we would be referred to a dietician and we haven’t heard neither hide nor hair from a dietician yet. I think it has been least a month now. It was recommended by the doctor and we said sure, yeh, anything right*?

Interestingly, one family member gauged the severity of the patient’s illness by what the health care provider did not implement by way of strategy, particularly the insertion of a feeding tube that she had seen done to treat a relative with an acute medical condition. This patient’s wife reasoned:

*I thought well, if it is getting that bad, they will know that he should be admitted and something should be done. If it isn’t bad enough that they aren’t recommending hospitalization and a feeding tube and stuff, maybe we are still safe for a little while yet. And maybe then I shouldn’t be so worried*.

*Chief Impacts/Fears:* When asked about the chief problems that lack of appetite and weight loss caused for themselves, family members described feelings of being chronically worried, frustrated, scared, and helpless. This patient’s wife explained:

*It’s just a constant worry about keeping that weight on and making sure that he eats. Its stressful, very stressful to see somebody that has been so vibrant change dramatically physically and mentally in front of you*.

The daughter of this patient shared:

*To see someone that you love so much going downhill and you are powerless to do anything about it. It’s depressing. You feel helpless*.

The waxing and waning of appetite and food cravings were challenging for families who were constantly striving to procure something that the patient would eat. This exasperated daughter shared:

*Her cravings drive me crazy so it is almost like a trip to the grocery store every day or every second day*.

The changes in physical appearance due to marked weight loss was emotionally difficult for family members. The husband of a patient stated:

*And when I see her unclothed and getting in the shower and stuff it just rips me apart to see her like that. It is like the people you see in the Nazi death camps. And I’m not kidding. It is just skin and bone*.

Images of holocaust victims was also invoked by the wife of a patient:


*It is scary. I saw how skinny he was getting. And I felt very helpless. He started to look like somebody that had gone through the Holocaust and it was just, he was just skin and bones.*


This wife believed that weight loss and muscle wasting experienced by her husband would be particularly traumatizing for men in general:

*I think it is almost like anorexia [nervosa] where you have image issues because you don’t see yourself as valuable. For me I think there is a parallel there. Like their body image for a man who is weak, I think they are reflecting on their self-worth. In a way anorexia is that way too because you want to become invisible because you are not projecting a good self-image*.

Having to assume new roles such as cooking, doing laundry, managing finances, and car maintenance were daunting and overwhelming for family members. These wives bemoaned:

*I feel stressed because I have to do everything now. Finding the time to do all the things that need to be done, because she is unable to do it. You have these other things to do after a long days work and the stuff is there and you have to suck it up and do it but it’s not easy. It is hard on a spouse when you are alone*.*When he was robust, he was very strong. But now he has no strength. If I couldn’t open a jar I would give it to him, right? And all of a sudden we changed roles. Now I am much stronger than he is and that is very hard*.

The data were equivocal regarding impacts on family members’ own eating habits. Some felt that they did not want to eat in front of their ill relatives. This patient’s son stated:

*I don’t eat in front of her at all. If she’s not eating, I’m not eating. So that’s kind of how it is*.

Others identified the importance of attending to their own nutritional needs. The wife of a patient explained:

*I try to maintain my own nutrition. Because I have to take care of myself. That is one thing that I have been very diligent about. I don’t enjoy it as much but I ensure that I do eat properly*.

Family members also spoke of missing out on the opportunities to dine together, host friends and families for meals, and overall decreased opportunities for socialization. These quotes are reflective of many family members’ experiences. This husband lamented:

*Oh yeah, we used to enjoy cooking together and well, we can’t do that now. It changes your relationship as a couple. She is often not at the table where the family is. And so you lose that eating together. Just doesn’t have the energy to get to the table. And we used to host a lot of people. We don’t do that anymore*.

This son observed:

*Her enjoyment of food is just not there. And she was always very social normally and always liked going out to eat. It was really a social thing and that is really gone*.

Oscillatory Wayfaring: This sub-theme reflects the impacts the patients’ cancer anorexia–cachexia has on family members and aligns with the portion of Kleinman’s explanatory model related to severity and prognosis of an illness and chief impacts and fears. Families oscillated between feeling optimistic and hopeful that patients would be able to gain weight and feeling resigned to what continuing anorexia and weight loss portended.

This family member understood that in her current condition, his wife was not strong enough for surgery to resect her tumor and hoped for improvement to allow for that intervention: 

*Well, we are just strengthening her body for the fight. We know it is a bad scene with her. So we hope the chemo is working but there may be some aggressive invasive surgery and so we know that she has to go in stronger. So basically keeping the body as strong as we can. That is my fear about the weight loss and appetite. That it gets so severe that organs start shutting down, if it gets medically dire kind of thing. I just wish there was this magic pill that they could give you when you are starving to death all of the time. The worst case scenario…death. You know, it all accumulating to the point where there is no return and it scares the living hell out of me*.

A patient’s wife who had previously pestered her husband to eat spoke of her change in approach as his condition deteriorated:

*At first, I was quite militant because I thought he was giving up. You have to eat to give your body strength. And when people stop choosing food, it is like they are stopping choosing life. And that was my anxiety for a long time. At one point you have to just step back and say, this is ridiculous. I don’t understand how he is feeling so I can’t club him over the head [about eating] which is what I was trying to do really. Now I try to approach it as I am taking or leaving it and giving him the choice. Now that we understand that the end is nearer than not a lot of things fall into place as far as what is important so that he had a good day, you know*?

The spouse of this patient described how anorexia and cachexia symptoms helped prepare her for the inevitable:

*It prepares you for what is happening. It is a very physical reminder every day that you are going towards decline and there is no pill to turn it around. A skeletal body prepares you for death. I’ve accepted that, so I don’t have fears about them anymore*.

## 4. Discussion

This grounded theory study examined the process by which family members arrive at their explanatory models of cancer anorexia–cachexia and explicated the points of convergence and divergence between family’s explanatory models and those in the biomedical literature. The core category, wayfaring, captures how families traverse the unfamiliar terrain of cancer cachexia to help them arrive at a semblance of understanding of and explanations about the patient’s illness. Early definitions of the concept of wayfaring found in the transportation literature were primarily concerned with moving through space with the goal of reaching a spatial destination [30,31,32,33]. Ingold’s more recent definition [34] characterizes it as an embodied experience of movement, which exists not within places but along pathways. As such, the wayfarer ‘…. is not place bound, but continually on the move’ (p. 35). Such unfolding movement was evident in this study as families moved from and through unrecognized, attributional, and oscillatory wayfaring.

The attributional wayfaring of family members aligns with social psychology literature exploring how people explain or assign attribution to the causes of events and the behavior of others [35]. In this study, some family members characterized their relatives as being picky eaters or just not trying hard enough to eat. This characterization is reflective of internal or dispositional attribution [36], where the cause of a behavior or event is related to a person’s internal characteristics such as personality, abilities, or effort. Families also made external attributions [37] to explain anorexia and cachexia, attributing them to such things as the side effects of treatment and the cancer diagnosis itself. During attributional wayfaring, families were also on other sojourns such as dealing with impacts related to the loss of communal dining with the patient, friends, and family, and undertaking new roles that the patient could no longer fulfill. The social and emotional impacts of cancer cachexia on family members emerging in this study are consistent with the findings of previous research [38,39,40].

In oscillatory wayfaring, family members moved between periods of hope for an improvement in their relative’s condition and sad resignation in response to the current and anticipated losses that would ultimately culminate in death. Hope in this study was not equated with cure but with improved appetite and weight gain, however small. Family members continued to deploy the strategies previously detailed in attributional wayfaring to achieve this end. This finding is consistent with Snyder’s work on hope that characterizes the construct as consisting of a goal, pathways or activities to achieve the goal, and a sense of agency that drives action forward [41]. The sad resignation dimension of oscillatory wayfaring is characteristic of what Singer and colleagues have described as pre-death grief, which consists of both illness-related grief and anticipatory grief [42]. The current and ongoing losses patients were experiencing during their illness were grieved by family members and were thus present-oriented. In contrast, anticipatory grief is characterized as the pending physical, and worry about the future after the patient had died is future-oriented.

### Family Explanatory Models and Biomedical Literature

Dinos and colleagues [43] observe that ‘…. explanatory models are not static entities or a single constructt, but can be fluid, multilayered and complex factors that may change in response to a number of factors……’ (p. 106). These characteristics are exemplified in the Wayfaring model reported in this study. Findings indicated that the explanatory models of CACS family members hold while consisting of a smattering of vaguely understood physiological causes, also containing perceptions and explanations about CACS that are not captured in biomedical understandings of this prevalent clinical problem. A summary of the points of divergence between family’s explanatory models and the biomedical literature is presented in Table 1.

Onset and Etiology: It was difficult for family members to pinpoint the onset of cancer anorexia–cachexia because patients’ appetites were variable from day to day, and weight loss was initially subtle. Appetite variation may be reflective of research findings linking food aversions and diurnal variations in appetite to early satiety of cancer patients [43]. Families found it particularly challenging to understand the early satiation patients experienced, sometimes attributing it to patients being stubborn or picky eaters. In contrast, the biomedical literature implicates a host of factors including the impact of gut hormones that contribute to delayed gastric emptying and distention [44]. Though families could readily identify emaciation in a relative, consensus definitions of cachexia characterize weight loss as occurring on a continuum [45]. Thus, in the early stages of disease, marked weight loss may not be evident. Moreover, research indicates that though detectable using diagnostic imaging, visible loss of weight and muscle mass may be masked in individuals with cancer who are overweight or obese [46].

Physiological Structures/Anatomical Processes Involved: Family members in this study identified that “the cancer” was causing patients’ weight loss and lack of appetite. Though not necessarily always able to elaborate on exact physical processes and anatomical structures involved, they thought tumors may be causing obstruction or stealing calories from the patient. Absent from their explanations were concepts of complex central nervous system and metabolic disruptions, elevated energy expenditure excess catabolism, or inflammation well documented in the biomedical literature [47,48,49,50,51]. 

A variety of symptoms were identified by family participants as impeding the patient’s appetite and ability to eat, including changes in taste and smell. Research confirms that compared to advanced cancer patients with no chemosensory disturbances, those with self-perceived severe chemosensory alterations consume fewer calories, lose more weight, and have poorer quality-of-life scores [52]. Whether due to illness or treatment, studies document that cancer patients often experience multiple symptoms occurring together. Research examining symptoms clusters in cancer patients conducted by Aktas and colleagues identified the cluster of fatigue/anorexia–cachexia/gastrointestinal [53]. The symptoms in this cluster of anorexia, early satiety, easy fatigue, lack of energy, nausea, taste change, vomiting, weakness, and weight loss are consistent with family reports of patients’ symptoms. 

The vilifying of morphine in causing anorexia identified in the study warrants comment. Well-documented gastrointestinal opioid-induced side effects include nausea, vomiting, and constipation [54]. While these symptoms were identified by family members, they were not specifically attributed to opioid medication. What was suggested was the stigmatizing belief that people receiving morphine do not eat because their addiction to medication supplants their desire for food. This finding underscores the need for family caregiver education regarding the metabolic drivers of anorexia and concepts of opioid dependence, tolerance, and addiction.

Treatment Options: Study participants attributed patient weight loss to a caloric deficit, reasoning that increased caloric intake results in weight gain. While intuitive, this reasoning does not hold in the setting of advanced cancer cachexia. Research has demonstrated that decreased caloric intake is not the sole driver of cachexia and implicates complex metabolic disturbances, increased energy expenditure, and marked catabolic activity [4,5]. Such perturbations explain the ineffectiveness of interventions such as total parenteral nutrition and tube-feeding to reverse cachexia. Accordingly, the American Society of Clinical Oncology (ASCO) Management of Cancer Cachexia Guidelines do not endorse the routine use of these interventions [13]. A lack of consensus on this point may fuel, in part, the conflict that can occur between health care providers and families about the appropriateness of nutritional intervention. Thus, exploration of family members’ explanatory models is an important step in tailoring family conversations about the role of artificial nutrition in the face of advanced disease.

Some family members believed that it was the responsibility of the patient to maintain sufficient caloric intake through sheer will if need be. Implicit within this belief is that patients who are ‘nutritional underachievers’ bear the responsibility for their physical decline which in turn may engender family member resentment. Grounded theory research conducted by Shragge and colleagues [55] examining psychosocial and dietary management of anorexia in advanced cancer patients identified the process of ‘shifting to conscious control’ whereby some patients retained the motivation to eat in the presence of anorexia as long as it did not trigger nausea. To the extent that it ameliorates the emotional and social impacts of anorexia, patients who wish to shift to conscious control should be supported in its use. An important caveat here is that family members do not force patients to eat when they do not feel like doing so.

Families in this study wondered about the efficacy of cannabinoids to manage CACS and believed it would be helpful earlier in the course of disease. However, a recent systematic review and meta-analysis conducted by Simon and colleagues concluded that evidence from both randomized controlled trials and non-randomized studies of interventions was of low quality, and that use of cannabinoids did not result in weight gain or improvements in appetite or quality of life [56]. Health care providers can anticipate that families may wish to explore the use of this medication as part of clinical care and must be prepared to respond with evidence.

Impact on Patient and Family: Participants’ explanatory models of cancer anorexia–cachexia contained reports of emotional impacts for patients and families. Body image is negatively impacted for patients, and previous research has identified that feeling self-conscious about marked weight loss and others’ reactions to it results in social isolation [57]. The marked wasting seen in patients with cachexia is associated with anxiety, depression, and diminished quality of life [58]. Female family study participants believed that weight loss and muscle wasting was particularly difficult for their male relatives, characterizing loss of strength as traumatizing for them. Psychological research examining gender differences related to negative body image indicates that typically, women are significantly more dissatisfied with their bodies than men with respect to excess weight [58]. However, consistent with beliefs of family members in this study, research indicates identity loss of muscularity and strength is problematic for men and may be a defining quality of their identity [59]. No specific research examining gender differences regarding the impact of wasting in the setting of cancer cachexia was located; however, research with HIV-infected men has reported that lipodystrophy is associated with poor body image [60].

It is interesting that the symptom of anorexia was characterized as an expression of suffering in this study. Such a characterization is akin to both classic [61] and recent scholarly work [62,63,64] about individual expressions of individual suffering and anxieties known as idioms of distress. Hinton and colleagues [65] note that the value of attending to idioms of distress in clinical work is that ‘… it may help clinicians understand sufferers’ view of the causes of their distress, constitute key therapeutic targets and help increase therapeutic empathy and treatment adherence’ (p. 209). 

The impacts of CACS on family members in this study are consistent with findings of studies conducted world-wide that have identified the caregiving experience as replete with losses, changes in role function, and feelings of distress and isolation [8,16,66,67]. There is increased recognition of the need to include a psychosocial component as part of a multimodal approach in caring for patients with cachexia and their family members. Hopkinson and colleagues have conducted seminal work in the development and evaluation of nurse-led multicomponent toolkits patients experiencing eating problems and involuntary weight loss [68,69] and postulate a synergistic effect between biomedical and psychosocial interventions in cachexia care [70]. Central to this is the ability of health care providers to educate patients and families about the condition.

## 5. Limitations

The findings of this study must be considered in light of its limitations. First, a single individual conducted the coding and data analysis, although it was an experienced grounded theory researcher. Birks and colleagues [71] note that ‘GT is often undertaken by individual researchers who carry out data collection/generation, analysis, and theory development autonomously’ (p. 3). It is acknowledged that analysis conducted by a team may have generated richer conceptualizations and explanations of the data. However, interpretations of the data were discussed and confirmed with the Research Nurse who conducted interviews with family participants. Second, the family members interviewed were those of patients with advanced cancer and marked cachexia. Explanatory models of cancer cachexia elicited from family members whose relatives are earlier in their illness may differ. Third, the majority of participants were female, and white Anglo-Saxon protestant. Thus, research examining explanatory models constructed by men and in diverse ethnic groups is warranted. 

## 6. Conclusions

Family members seek to make sense of the anorexia and cachexia experienced in the setting of advanced disease through constructing explanatory models using the process of wayfaring. Their responses to the patient and health care providers are shaped by notions in explanatory models related to etiology, treatment, and consequences/impacts. Findings from this study suggest a divergence between family understanding about the causes and management of CACS and biomedical literature guiding practice. Discussions with families to elicit their explanatory models of CACS are important for clinicians in order to better understand the wider belief systems that are operating in the clinical encounter and provide the foundation for formulating and implementing treatment plans.

## Figures and Tables

**Figure 1 healthcare-12-01610-f001:**
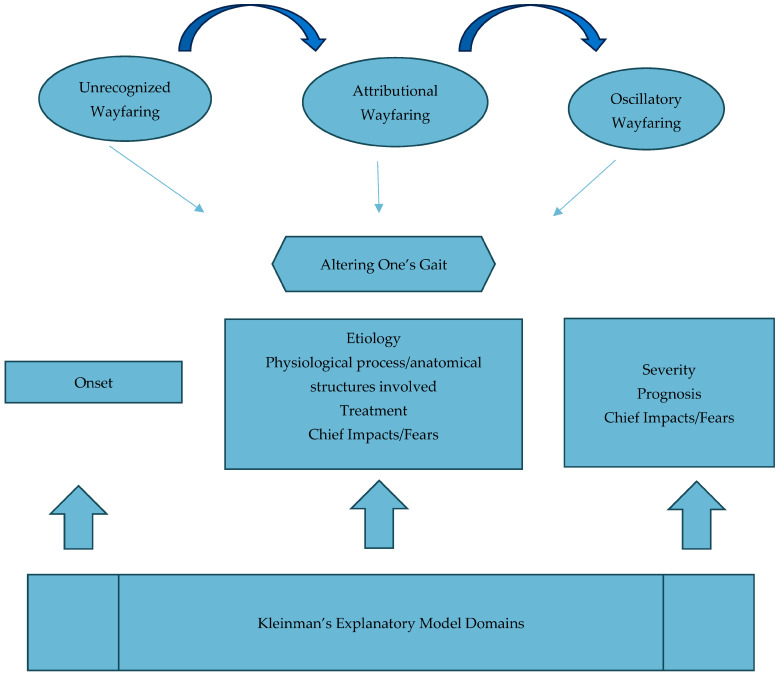
Wayfaring Model and Relationship to Kleinman’s Explanatory Model.

**Table 1 healthcare-12-01610-t001:** Points of divergence between family’s explanatory models of CACS and the biomedical literature.

	Family Member Explanatory Models	Biomedical Literature
Etiology of Anorexia and Cachexia	Patient-related factors: ‘picky eater’ ‘being stubborn’ Disease-related factors: tumor stealing calories tumor ‘plugging things up’ Treatment-related factors being on morphine akin to being an addict and ‘addicts don’t eat’	Gut hormones contribute to delayed gastric emptying and early satiety Central nervous system and metabolic disruptions cause increased energy, catabolism, and inflammation Opioids may cause nausea and vomiting, but do not drive anorexia
Treatment of CACS	Increase caloric intake e.g., implement tube feeding prescribe cannabinoids to stimulate appetite	American Society of Clinical Oncology (ASCO) Management of Cancer Cachexia Guidelines does not endorse the routine use of tube feeding for advanced cancer patients outside the context of a clinical trial Randomized controlled trials and non-randomized studies, though of low quality, concluded cannabinoids did not result in weight gain or improvements in appetite or quality of life

## Data Availability

Data are unavailable due to restrictions imposed by the University Ethical Review Board at the time the study received ethical approval.

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
