# Peer review of "Family Members’ Explanatory Models of Cancer Anorexia–Cachexia"

_healthcare, 2024, doi:10.3390/healthcare12161610_

Round 1

Reviewer 1 Report

Comments and Suggestions for Authors

The authors contend that the use of Explanatory Models will improve clinician insight on patient family's belief regarding their CACS.  The authors provide a compelling argument with evidence that family feedback via EMs may improve family-clinician interaction by improving the clinicians knowledge of the families understanding of CACS but, also assist in getting family, clinician, and patient on the same page regarding treatment intent and reasoning. 

Minor Concerns

Line 47-48 – grammatical error

Line 54 – punctuation

Line 59-60 – need citation.

Line 74-75 consider rewording.

Line 424 – the word “with” needs to be added

Introduction

Authors state “Variation in family members’ explanatory models of illness and those articulated in the biomedical literature, while perhaps reflecting some common underlying understandings may have important dimensional differences that ground family-health care provider conflicts.” Without providing explanation/evidence (with citations) and go on to base their research questions on this premise. Authors should build a case for posing these research questions. What specific evidence or gap in knowledge led to these research questions? This information is needed to provide a premise for the research questions. Although detailed in the discussion, a definition that distinguishes one from the other should be included in the introduction.

Methods

The validation tool used to define cachexia was developed in 2002 (PG-SGA). The definition has since been updated and published in a {Fearon, 2011 Definition and classification of cancer cachexia: an international consensus}.  Authors should describe why the outdated definition was used.

Results 

How were the statements made by family members chosen to be included in the article?

No reporting of family members educational level or background was included. Might some of these opinions be different if family members have a some education in health and human physiology?

Author Response

Dear Reviewer #1: Thank you for your careful review of this manuscript and thoughtful feedback provided. Responses are provided below.

  • The authors contend that the use of Explanatory Models will improve clinician insight on patient family's belief regarding their CACS.  The authors provide a compelling argument with evidence that family feedback via EMs may improve family-clinician interaction by improving the clinicians knowledge of the families understanding of CACS but, also assist in getting family, clinician, and patient on the same page regarding treatment intent and reasoning. 

Response: Thank you

  • Minor Concerns

Line 47-48 – grammatical error (Now lines 54-58 in revised manuscript)

Response: This is the text the reviewer is commenting upon (lines 47 and 48  in the original version of the manuscript (see below). I have re-read carefully and cannot find any grammatical errors. I have also run a check for grammar with my word processing program and no grammatical issues were identified. Perhaps the reviewer was referring to different lines in the manuscript? If so, please advise, and I will be happy to correct.

While studies have been conducted detailing what family members do in response do in response to the symptoms of CACS [14-16], empirical work systematically examining why family members do what they do is lacking. Thus research is needed to provide a comprehensive understanding of the ways that family members make sense of CACS in the context of advanced cancer.

Line 54 – punctuation (Now line 61 in revised manuscript)

Response: Thank you for catching this. A period has been added to the end of this sentence.

            “……about a particular episode of illness and its symptoms. [17]

Line 59-60 – need citation. (Now line 67 in the revised manuscript)

Response: A citation has been added as requested. A limitation of the CSM is that it fails to explore notions of cure or control of symptoms. [19]

Line 74-75 consider rewording. (Now lines 79-82 in the revised manuscript)

Response: Reworded as requested:

Revised wording: Research explicating an understanding of family members’ explanatory models of CACS will sensitize clinicians to potential areas of divergence. This understanding will allow for the development of interventions responsive to family members’ unique needs.

Line 424 – the word “with” needs to be added (Now line 454 in revised manuscript)

Response: Thank you for catching this. The word has been added so that the sentence now reads:

“…….During attributional wayfaring families were also on other sojourns such as dealing with impacts….”

Introduction

Authors state “Variation in family members’ explanatory models of illness and those articulated in the biomedical literature, while perhaps reflecting some common underlying understandings may have important dimensional differences that ground family-health care provider conflicts.” 

Without providing explanation/evidence (with citations) and go on to base their research questions on this premise. Authors should build a case for posing these research questions. What specific evidence or gap in knowledge led to these research questions? This information is needed to provide a premise for the research questions. Although detailed in the discussion, a definition that distinguishes one from the other should be included in the introduction.

Response: Thank you for this feedback. Information has been added to provide an explanation and evidence with citations to substantiate the research questions that were posed. Please see lines 42-52.

Methods

The validation tool used to define cachexia was developed in 2002 (PG-SGA). The definition has since been updated and published in a {Fearon, 2011 Definition and classification of cancer cachexia: an international consensus}.  Authors should describe why the outdated definition was used.

Response: Thank you. Indeed, we are aware of the 2011 definition developed by Dr. Kenneth Fearon and colleagues published in Lancet Oncology. The decision to use the PG-SGA for the study was driven by a few factors. First, reviewer feedback from the palliative care panel of the funding body we applied to requested it’s inclusion in a resubmission for funding. Second, it is a validated method of nutritional assessment based on the featured of medical history, weight change, dietary intake, change, gastrointestinal symptoms. Thus we were confident that the tool would identify patients with CACS who would then nominate family member participants—the primary informants for our study.

Results 

  1. How were the statements made by family members chosen to be included in the article?

Response: Thank you for this important question. Far beyond just including data exemplars because it is customary to do so in reporting qualitative research, decisions about which family statements to include were informed by the author asking herself the following questions: Which statements best reveal the participants’ emotions and experiences? Which statements evoke emotion, or provoke response in the reader? Do the statements presented provide the reader with enough representative data to support the findings and any alternative explanations? Do the family comments included represent a careful selection of the complete data set, sustain a descriptive analysis, and enable the reader to enter the lifeworld of the family participants? Do the statements provided illustrate for the reader how the presented findings and interpretations have arisen from the data?

No reporting of family members educational level or background was included. Might some of these opinions be different if family members have a some education in health and human physiology?

Response: Additional information regarding family participants’ characteristics, including educational level/background have been added to the manuscript (Please see lines 179-184).

Family participants in this study were, overall, quite well educated. Eight family members had completed a university degree; 10 had completed some community college education; 7 had completed high school.  There were no discernable differences between the explanatory views held between those with different levels of education.  It is possible that had a physiologist with a background in cachexia research or palliative care clinician been a study participant that their opinions would be different. And, while intuitive and reasonable to consider that a more advanced educational background and/or health care background might result in explanations more aligned with the prevailing medical model, this was not borne out in this study, nor in the author’s previous research wherein palliative care clinicians indicated that their medical colleagues in other domains of medical practice were ill informed about the etiology and management of cancer cachexia. Extant literature also consistently underscores the need for increased awareness among health care providers about the etiology of CACS and its management.  

Reviewer 2 Report

Comments and Suggestions for Authors

I very much enjoyed reading this well-written paper exploring how family members of people with cancer cachexia develop explanatory models of cancer cachexia and how these models differ from the biomedical evidence.

Introduction

The introduction is very clear and sets out the rationale for the study well. Research question one focusses on the process by which family members arrive at their explanatory models. Based on the introduction, I was expecting a research question to cover what the explanatory models actually are, in addition to the process. Perhaps that is implicit, but it would be good to make it more explicit.

Method

I am not an expert in grounded theory but my understanding is that it is a very open approach. This does not seem to quite match the use of the Kleinman questions, suggesting a more structured approach, so this needs addressing. Related to this is that the interviews were carried out by a study nurse, who is not an author on the paper. It seems strange not to include the study nurse as an author when they had such an integral role in the study, especially if the interviews were open and relatively flexible, which I think would be most people’s expectation for a grounded theory study. I think there need to be a bit of clarity around the role of the study nurse in this project – and also how many were involved as it’s not clear whether there was one or more.

Please make clear in the methods that the analysis was carried out by one person. This is noted as a limitation in the discussion section, but it would be good to say a little bit more here about why no-one else was involved (why not the interviewer for example?) and how this could impact the findings. In most qualitative research studies, there is a team of people involved, although I know this is not always the case in grounded theory so a reference to this would be beneficial.

Results

I would expect a fuller description of the participants – perhaps a table could be included to summarise this.

Would it be possible to label the quotes in some way to provide some context e.g. daughter, wife? This would also make it clear if quotes are coming from different participants.   The narrative appears to be missing for the quotes on lines 266-270.

Much of the content in the Attributional Wayfaring section doesn’t seem to fit that category as it seems to be more about experiences than explanations and it’s not clear how this contributes to the research questions (Lines 266-270, 276-372). Similarly, some of the content in the Oscillatory Wayfaring seems to be more about the experience. It may be that a little more narrative is required around the quotes to highlight how this addresses the research questions.  

There are some formatting issues with Figure 1.

Discussion

Line 421 – I have not come across the term ‘international characteristics’ before – is that correct?

Line 434 – Snyder typo

The introduction highlighted that by understanding the points of divergence between family’s explanatory models and the biomedical literature, clinicians will be in a better position to provide support. To facilitate this, a table summarising the points covered in the discussion related to this would be really helpful.

Author Response

Dear Reviewer #2: Thank you for your careful review of this manuscript and thoughtful feedback provided. Responses are provided below.

I very much enjoyed reading this well-written paper exploring how family members of people with cancer cachexia develop explanatory models of cancer cachexia and how these models differ from the biomedical evidence.

Response: Thanks very much!

  • Introduction

The introduction is very clear and sets out the rationale for the study well. Research question one focusses on the process by which family members arrive at their explanatory models. Based on the introduction, I was expecting a research question to cover what the explanatory models actually are, in addition to the process. Perhaps that is implicit, but it would be good to make it more explicit.

Response: Thank you for this helpful suggestion for revision. An explicit statement has been added to the manuscript. See lines 474-481.  

2)      Method                                                                                                                                                 I am not an expert in grounded theory but my understanding is that it is a very open approach. This does not seem to quite match the use of the Kleinman questions, suggesting a more structured approach, so this needs addressing. Related to this is that the interviews were carried out by a study nurse, who is not an author on the paper. It seems strange not to include the study nurse as an author when they had such an integral role in the study, especially if the interviews were open and relatively flexible, which I think would be most people’s expectation for a grounded theory study. I think there need to be a bit of clarity around the role of the study nurse in this project – and also how many were involved as it’s not clear whether there was one or more.

Response:  The reviewer is correct that in classic grounded theory work that only the information collected by the gathered data should influence the progress of the research and that there are no pre-set questions prior to data collection. The method used in this project is more aligned with a constructivist grounded theory approach wherein the researcher constructs rather than discovers, and the research questions influence how data is collected. This approach does not preclude the use of semi-structured interview guides, such as the one by Kleinman used in this study. Conlon and colleagues (2015)  note that semi-structured interviews are more suitable for a grounded theory study when the researcher has identified, albeit tentatively, some domains that have already situated the inquiry which interviewing can then begin to expand upon. The expansion involved the researcher asking questions about the core category of wayfaring that emerged. For example, asking the question, When does wayfaring revealed the unfolding and temporal nature of the process during the patient’s illness.  Asking the question,  How does wayfaring occur, helped to reveal the different facets of experience seen in unrecognized, attributional, and oscillatory wayfaring.

Conlon C., Carney G., Timonen V., Scharf T. (2015). “Emergent reconstruction” in grounded theory: Learning from team-based interview research. Qualitative Research, 15(1), 39–56. https://doi.org/10.1177/1468794113495038

       3)    Please make clear in the methods that the analysis was carried out by one person. This is noted as a limitation in the discussion section, but it would be good to say a little bit more here about why no-one else was involved (why not the interviewer for example?) and how this could impact the findings. In most qualitative research studies, there is a team of people involved, although I know this is not always the case in grounded theory so a reference to this would be beneficial.

Response: Thank you for the chance to respond to this. I recognize that having one analyst on a grounded theory study is far from ideal and was not the original intent when embarking on this study. Apart from my own grounded theory dissertation work nearly two decades ago, my subsequent program of research has always engaged a team in the analysis of grounded theory work. This is the only grounded theory study I have conducted in which I was the primary analyst. The reason for this is a sad one.  Due to issues of sensitivity, confidentiality, and the desire to protect anonymity, I cannot elaborate on in the manuscript, but will explain here. The colleague with whom I had planned to collaborate on the analysis suffered a severe mental health breakdown during the project and has not been able to return to work. It was important to me to honor the contributions that family members had made to this project, and disseminate the findings as is the expectation of the funding body. Thus, rather than let the data remain unanalyzed and unpublished, given my experience and facility in conducting grounded theory research, I made the decision to go forward and conduct the analysis independently.

While optimally conducted in a team, there is commentary in the literature about the solo nature of grounded theory work, particularly if the researcher is experienced in the grounded theory method. For example, see Birks and colleagues who note that, “GT is often undertaken by individual researchers who carry out data collection/generation, analysis, and theory development autonomously” (p. 3). [Birks, M., Hoare, K., & Mills, J. (2019). Grounded Theory: The FAQs. International Journal of Qualitative Methods18.                                                                                                                               https://doi.org/10.1177/1609406919882535]

I have elaborated on the issue of a sole analyst in the Limitations section of the manuscript.  Please see lines 588-595.

Two research nurses experienced in conducting qualitative interviews collected the data for this study. I have added a statement in the manuscript to reflect this. Please see line 134. Neither were  involved in the work of coding as it was not our practice in the Research Unit where I worked at the time to budget for their time to participate in this activity.  However, appreciating their role in the co-constructing the data captured in the interview, and as part of supervising her work, I would have regular conversations with them and ask questions about the data to verify and ensure that I was understanding the nature and context of a participant’s response. This has also been added to the manuscript. Please see lines 595-597.

  • Results I would expect a fuller description of the participants – perhaps a table could be included to summarise this.

Response: More descriptive data regarding the characteristics of the family members participating in the study has been added to the text of the  manuscript. Please see lines 179-184.

Would it be possible to label the quotes in some way to provide some context e.g. daughter, wife? This would also make it clear if quotes are coming from different participants.   The narrative appears to be missing for the quotes on lines 266-270.

Response: Thank you for this good suggestion. Information has been added following each of the data exemplars to provide context for the reader. Missing narrative has been added.

  • Much of the content in the Attributional Wayfaring section doesn’t seem to fit that category as it seems to be more about experiences than explanations and it’s not clear how this contributes to the research questions. Similarly, some of the content in the Oscillatory Wayfaring seems to be more about the experience. It may be that a little more narrative is required around the quotes to highlight how this addresses the research questions.  

Response:  Data exemplars that do not speak to attribution have been removed from the Attributional Wayfaring section. Data exemplars that do not speak to experience have been removed from Oscillatory Wayfaring.

  • There are some formatting issues with Figure 1.

Response: I am wondering if this is a function of uploading the figure into the editorial system as the document that was originally submitted did not have the formatting issue identified. I will upload a new figure to see if that rectifies things.

  • Discussion

Line 421 – I have not come across the term ‘international characteristics’ before – is that correct?

Response: Thanks for catching this. The correct wording is ‘internal characteristics’, and has been added to the manuscript. Please see line 451.

Line 434 – Snyder typo

Response: Thank you for catching this. It has been corrected. Please see line 464.

  • The introduction highlighted that by understanding the points of divergence between family’s explanatory models and the biomedical literature, clinicians will be in a better position to provide support. To facilitate this, a table summarising the points covered in the discussion related to this would be really helpful. Response:  I appreciate this good suggestion. A table summarizing the points in the discussion has been constructed and added. See Table 1.

Reviewer 3 Report

Comments and Suggestions for Authors

Abstract: To add more results ref to data collection

Introduction: What is main objective of study? What is hypothesis?

Methods: What are the questionnaire used to data collection? It is validated?

82 patients were included. But, what minimum necessary to understand the relationship cancer-cachexia anorexia?

Results:

This manuscript is very descriptive. Once using the cachexia and anorexia the data regarding to body mass, body mass loss, fat mass, BMI, food intake or appetite is crucial to understand the data collected.

Author Response

Dear Reviewer #3: Thank you for reviewing this manuscript and providing feedback. Here are the responses to the items raised:

  • Abstract: To add more results ref to data collection

Response:     It is not exactly clear from this brief comment what in particular with respect to the results the reviewer wished to see added. A synopsis of the overarching findings are articulated in the abstract. The work by Kleinman is appropriate referenced when cited in text in the body of the manuscript.

  • Introduction: What is main objective of study? What is hypothesis?

Response:     There is a clear statement on page 2, lines 84-87 that articulate the research questions driving the study. The study that was conducted was not hypothesis generating in its aim, thus statement of a research hypothesis was not included as it would not have been appropriate.

  • Methods: What are the questionnaire used to data collection? Is it validated?

Response:     Questions of validation of study questionnaires are appropriate for quantitative research. However, this was a qualitative research study. In qualitative work  interview guides are used to structure the conversation between the study participant and data collector. In this study, the interview guide was informed by Kleinman’s explanatory model framework questions. Interview guides are not subject to psychometric evaluation, as they are not purporting to measure anything. Thus psychometric properties are not reported.

  • 82 patients were included. But, what minimum necessary to understand the relationship cancer-cachexia anorexia?

Response:     In a non-positivist paradigm, sample sizes are not calculated a priori as they are in quantitative studies. Rather, sample size is determined by the richness and fullness of the data and the extent to which saturation occurs—that point in data collection where no new findings emerge and participant reports continue to confirm existing data rather introduce varied exemplars of the phenomenon of interest. Sample sizes tend to be smaller in qualitative work than in quantitative studies, and published reports may have anywhere between 10-50 participants—again, depending on the point where saturation is reached. This can’t be determined in advance because it depends on the quality of data generated in each interview.

  • This manuscript is very descriptive. Once using the cachexia and anorexia the data regarding to body mass, body mass loss, fat mass, BMI, food intake or appetite is crucial to understand the data collected.

Response:     The manuscript is indeed descriptive, consistent with the qualitative nature and design of the project. It was not the goal of the study to correlate the explanatory models generated with anthropomorphic measures. The PG-SGA was used solely for screening to ensure patients met a standardized measure of CACS, so that family members who were subsequently interviewed were all providing data about their experiences with a similar clinical phenomenon.

Round 2

Reviewer 2 Report

Comments and Suggestions for Authors

Thank you for addressing the comments in my original review. I appreciate you sharing the information about your colleague and I'm sorry to hear about their health difficulties.

Author Response

I appreciate your comment about this. It has been devastating to watch an esteemed colleague suffer in this way.

Reviewer 3 Report

Comments and Suggestions for Authors

Too small sample size.

Author Response

As the study has already been completed, this issue cannot be addressed. More to the point, however, is that in qualitative work, large random sample sizes such as those that are used in quantitative work are not the norm. The issue of saturation, the point at which no new information is forthcoming is the criterion by which adequacy of sample size is determined using a non-positivist approach.